# Important Synthesis Parameters Affecting Crystallization of Zeolite T: A Review

**DOI:** 10.3390/ma14112890

**Published:** 2021-05-28

**Authors:** Siti Z. Patuwan, Sazmal E. Arshad

**Affiliations:** Faculty of Science and Natural Resources, Universiti Malaysia Sabah, 88450 Kota Kinabalu, Sabah, Malaysia; sitizubaidahpatuwan@gmail.com

**Keywords:** zeolite T, offretite, erionite, hydrothermal, microwave, secondary growth

## Abstract

Synthesis of zeolite T with a variety of desired characteristics necessitates extensive work in formulation and practical experiments, either by conventional hydrothermal methods or aided with different approaches and synthesis techniques, such as secondary growth or microwave irradiation. The objectives of this review are to adduce the potential work in zeolite T (Offretite-Erionite) synthesis, evaluating determining factors affecting the synthesis and quality of zeolite T crystals. Attention is given to extensive studies that interconnect with other significant findings.

## 1. Introduction

Zeolites are well-known porous crystal systems that exist in various sizes (nano-, micro-, etc.) and have been widely incorporated in different applications. They consist of a uniform size of pore distributions [1] which are also known as voids. Voids and cavities within the zeolite system are the factors that contribute to the unique properties of the zeolite. These voids and cavities are controlled by the type of crystals that grow and intergrowth within the zeolite membrane. Introduced for the first time by Bennet and Grad [2], nanoporous zeolite T (the rod-like shape shown in Figure 1) is identified to have within it an intergrowth framework that makes zeolite type T an exceptional species in the zeolite family.

This intergrowth system is made up of a cancrinite group of hexagonal crystals of offretite (OFF) and erionite (ERI) (Figure 2). The interframe of these two entities causes an alteration in the crystal arrangement of zeolite T, which is AABAACAABAAC, instead of AABAABAAB [4]. Despite stacking within the zeolite system, OFF-ERI gives characters to zeolite T in terms of thermal stability, mechanical and chemical stability [1], acid resistance, and high ion selectivity [5], which makes zeolite T a promising candidate in catalysis, molecular sieves, ion exchange, and other applications [6].

Unlike other zeolite members, zeolite T possesses different void sizes due to offretite and erionite intergrowth group. Offretite has 12-ring channels parallel to the c-axis and additional 8-ring subchannels that are normal to the c-axis. The pore size for the 12-ring channels is 0.67 nm × 0.68 nm, while 0.36 nm × 0.49 nm is the pore size for the 8-ring subchannels. Alternatively, erionite only has 8-ring channels parallel to the c-axis, with a pore diameter of 0.36 nm × 0.51 nm [1]. Generally, the framework of OFF and ERI may differ, but they are closely related. Notably, the ionic behavior of pure OFF and ERI is similar to the synthetic intergrowth, offretite-erionite, in zeolite T. Thus, zeolite Ts can be considered as intergrowth phases and are homogenous mixtures of two component phases of offretite and erionite [4,7].

### 1.1. Hydrothermal Conventional Method

Hydrothermal treatment is the most conventional and widely applied method for synthesizing zeolite T and is considered to be the earliest synthesis method, leading to high percentage (88–90%) product yields. It is a treatment that involves high temperature and pressure in the presence of water that acts not only as a solvent, but also aids in transferring pressure. It is a reaction mediator, capable of altering the physical and chemical properties of reagents or products, and it possesses the potential to be an accelerator in chemical reactions [8,9,10]. The hydrothermal synthesis technique is usually conducted on a mixture in a sealed vessel, where heat treatment is applied. This procedure is also labeled as an in situ reaction that strongly depends on temperature and time [1]. This method is easy to control during the synthesis phase, and recently, via hydrothermal treatment, the high purity and crystallinity of zeolite T were synthesized in 100 °C for 168 h [11,12].

Additionally, the synthesis method of zeolite T has evolved with the emergence of other methods fused with hydrothermal treatment, such as (i) microwave-assisted [9,10,11,12], (ii) sonochemical-assisted, and (iii) secondary growth techniques, for the development of zeolite T membrane.

### 1.2. Microwave-Assisted Method

The microwave-assisted method has been widely applied in zeolite synthesis for a decade due to its capability to rapidly produce high purity products with less time consumption [9]. With microwave assistance, the zeolite T membrane’s preparation time was shortened by 12 times, to only 9 h [3,11]. 

The effect of microwave irradiation gives impetus to the annihilation of normal hydrogen (H) bonding in water, causing ion oscillation and dipole rotation, which produces isolated, so-called active water molecules, which respond to the high rate of dissolving of the composition gel. Coupling the behavior of the microwave with the hydration of shell water molecules is efficient and favors the coordination change of the alumina octahedral layer to the tetrahedral layer, which is more reactive and essential in growing zeolite T materials. Hence, it results in thinner membranes that affect zeolite T’s permeability properties [10,11].

Figure 3 shows the morphology of thin, compact zeolite T membranes. Growing the rod-like zeolite T upon the alumina support tube required a secondary growth technique and several heating steps. The heating process affects the distribution of zeolite T on the membrane, and the distribution affects the membrane selectivity performance. It was found that the low power of microwave heating (4.8 kW/m^3^) led to insufficient growth of zeolite T (Figure 3a), while as the power density increased to 6.0 kW/m^3^, the membrane evolved with more defects (Figure 3b). Therefore, by controlling the heating power density of the microwave, high-performance zeolite T membranes can be synthesized [11]. 

### 1.3. Sonochemical-Assisted Method 

The sonochemical-assisted method uses powerful ultrasound radiation (20 kHz–10 MHz) to treat the homogenous mixture before hydrothermal treatment. Nucleation and formation of zeolite T have been superior due to sonochemical effects from acoustic frequency usage, and this method has shortened the synthesis period from 168 h to 24 h [13]. Figure 4 shows the morphology of zeolite T synthesized with and without sonication pre-treatment. 

The sonochemical-assisted method incorporates a similar concept to the microwave-assisted method, where the effect of vibrations influences the physicochemical phenomena that impute the nucleation and crystallization of zeolite T. In contrast to the microwave-assisted method, however, the sonochemical-assisted method stimulates a chemical reaction between solid and liquid in the zeolite T mixture by producing high energy acoustic cavitation, which will lead to the production of extremely high temperature (>5000 °C), pressure (>20 MPa), and very high cooling rates (>107 K s^−1^) [14].

### 1.4. Secondary Growth/Dip-Coating

The secondary method is a combination of primary method, where zeolite T is synthesized by following the optimum formulation. It is repeated twice using a supporter membrane (organic or inorganic), allowing more zeolite crystals to grow within the first system in two or more different composition gel formulations. Secondary growth is always related with seeding zeolite crystal seed on the supports, in direct hydrothermal in situ crystallization, either in a clear solution or in an aqueous gel inside an autoclave at a different temperature, depending on the types of zeolite; 90–100 °C for faujasite (FAU) or Linde type-A (LTA) while 180 °C for MFI type membranes [15]. The standard method for secondary growth is dip-coating, where the support base is dipped into the solution mixture containing the seed or crystal of zeolite T. 

Figure 5 shows the illustrated dip-coating techniques’ execution. Using this technique gives an advantage in controlling the desired thickness of a membrane.

Consequently, Figure 6 shows the morphology of the zeolite T membrane prepared in different coating layers. Its thickness controls the selectivity and permeability properties of the membrane. As the coating cycle increases, the membrane terrace’s thickness and hence, the gas’ ideal selectivity is improved, while the gas’ permeance is reduced [9,16]. 

Despite the flexibility of the seeding method, several factors need close attention to maintain the quality and purity of the zeolite T crystal that grows on the support. Defects may occur during the process of growing the zeolite T particles, and the combination of this technique with microwave treatment helps to form a uniform oriented zeolite T membrane [11]. 

Moreover, (i) the surface roughness of the support, (ii) suspension concentration, (iii) seed size, and (iv) coating temperature are essential factors that will affect the coverage, thickness, and defect-free property of the dip-coating technique [15]. As the zeolite T seeds grow on the support surface, the seeds’ distribution is controlled by the suspension concentration, which is attributed to the layer thickness. Eventually, the thick membrane will lead to the formation of microcracks during drying and curing, and ultimately affect the quality of the membranes [9,17,18]. 

Additionally, the temperature for drying and curing the membrane also plays a vital role in the prepared membrane’s quality. Figure 7 shows the morphology of the zeolite T membrane using a SiC polished tubular mullite support (outer diameter = 12 mm, thickness = 1.5 mm, and average pore of 1.0 µm) that has undergone the drying and curing process at temperatures ~100 °C and ~150 °C, respectively [5]. As the drying and curing temperature increased to 150 °C, the top layer’s intensity is more robust than the membrane prepared at 100 °C. 

## 2. Factors Affecting Crystallization of Zeolite T

Zeolite T can be prepared using various ways, either using independent chemicals or aided with seeding support. The only matter in synthesizing zeolite T is during the crystallization phase, which determines the effectiveness of the preparation requirements. Meanwhile, crystal formation of zeolite T (either by hydrothermal synthesis or aided with microwave-, or sonochemical- irradiation) is affected by several primary factors. From the past and recent literature, several cases involve crystals’ behavior upon several factors, namely SiO_2_/Al_2_O_3_ ratio, alkalinity (n (OH)–)/n (SiO_2_)), water ratio, synthesis temperature and crystallization time, and the addition of structure-directing agents (SDA) [1,3,4,8,16,19,20,21,22], while pressure is not directly involved in zeolite T synthesis [23]. This review summarizes the factors that affect the growth of zeolite T. 

### 2.1. Effects of SiO_2_/Al_2_O_3_ Ratio

Zeolite T is classified as intermediate silica zeolites (SiO_2_/Al_2_O_3_ composition ratio from 2 to 5 [22,23,24], hence the ratio of SiO_2_/Al_2_O_3_ will affect the crystallization behavior of zeolite T. Percentage yield of zeolite T can be maximized by increasing the molar ratio of SiO_2_/Al_2_O_3_ in the mixture. However, the optimum ratio for zeolite T is in the range of 3 to 4 [1,2,3,4,5,6,9,10,11,13,15,20,21,22,24]. 

However, the ratio of SiO_2_/Al_2_O_3_ depends on the alkalinity of the composition. Since zeolite T favors high alkalinity crystallization, controlling the ratio of SiO_2_/Al_2_O_3_ is crucial to prevent it from exceeding the optimal range (3 to 4). Otherwise, the composition mixture can promote the crystallization of competitive phases, such as zeolite L and/or W [2,20]. The SiO_2_/Al_2_O_3_ ratio was found to affect the crystallization behavior, such as nucleation rate, crystal purity and intensity, and application performance [1,2,4,5].

The amount of silica used to synthesize zeolite T, being higher than the amount of alumina, contributes to the stability characteristics in both thermal and acid, while remaining hydrophilic [11,25]. Hence, the ratio of SiO_2_/Al_2_O_3_ not only affects the crystallization behavior but also the application performances of the synthesized zeolite T.

### 2.2. Effects of Alkalinity

Zeolite T is synthesized under activation of the alkaline medium, which is usually made up of mineralizer components of Na_2_O and K_2_O. The alkalinity strongly influences the level of crystallinity and stability of the zeolites structures, thus affecting the performance of zeolite T in its applications. Generally, when the medium is in high relative alkalinity at about 0.71 [2], it will accelerate the nucleation of zeolite T crystals and shorten the crystallization time, because at a high pH, Na_2_O and K_2_O tend to loosen and dissolve by releasing their OH–, and this accelerates the reaction between aluminate and silicate ions, while the degree of silicate anions polymerization decreases [13,26].

However, at higher relative alkalinity beyond 0.71 [2], synthesized crystals of T-zeolite tend to redissolve, resulting in a low-density product, affecting the performance properties of zeolite T [13]. This phenomenon is due to the characteristic of aluminosilicate layers, which tend to aggregate, forming another phase in alkalinity ≥ 0.82 [27]. This study proved that the relative alkalinity has a substantial impact on zeolite T crystallization [2]. Different relative alkalinity yields different purity of the product; as the value increases, the zeolite T crystals’ growth was affected. Table 1 and Figure 8 elucidate the product formed from different zeolite species due to different relative alkalinity values. Meanwhile, the evolution phase of zeolite T crystals and the morphologies for each sample are shown in Figure 8a,b.

Thus, the synthesis of zeolite T must be in between a relative alkalinity of 0.71 [1,2] held only at optimum crystallization time (24–168 h) to allow the growth inclination. As the relative alkalinity increases, it will trigger the evolution of species zeolite L and W.

### 2.3. Effects of pH and Water Ratio

The water component added into the zeolite T system is calculated and presented explicitly in an adequate amount, as the medium’s alkalinity depends on the amount of water added. Zeolite formation is greatly affected by the pH values range (0.71–0.82), which corresponds to the values of water precursors of the solutions [3]. Thus, the pH of the alkaline medium can be varied by altering the amount of water in moles of 8–16 fused in. When the amount of water inside the system decreases, it initiates the mineralizer from Na_2_O and K_2_O to release OH-, causing the pH and alkalinity to increase [3]. As the synthetic medium is high in alkalinity, zeolite T particles’ size decreases because zeolite T’s aggregation size decreases as the water component decreases [27]. Moreover, the mixture’s water content will reduce the degree of OH– concentration, leading to decreasing monotonic zeolite growth and crystallinity [28]. 

Unfortunately, crystallization of zeolite T will be stunted if the relative amount of water is lower than 14 H_2_O, as the synthetic mixture’s reagent’s chemical reaction is incomplete. This is due to the function of water in the system as an ion transporter, also known as cavities. However, an excessive amount of water needed in the system will diminish the zeolite T crystals into an amorphous system [19]. Thus, an excessive amount of water inside the synthetic solution will cause poor selectivity of the zeolite T crystals intergrowth layer. Meanwhile, Figure 9 shows the XRD analysis for samples synthesized at water, mole 25, with no zeolite crystal form and indicates the amorphous phase evolved. Therefore, only an adequate mount of water, mole 14–16 H_2_O, is needed in performing the optimum crystallization of zeolite T, as the water controls the pH ranges [2,3].

### 2.4. Effects of Synthesis Temperature and Crystallization Time

Generally, temperature and time have a crucial impact upon T-type zeolites during crystallization. Although zeolite T synthesis is aided with different nucleation agents, the crystals phase may develop defects that are observable in other zeolite phases and species if the composition gel is exposed to insufficient or excessive temperature, or inappropriate crystallization time. Meanwhile, the authors [1] studied the effect of crystallization temperature time in synthesizing zeolite T and parsed Ostwald’s theory, where under the same synthesizing condition, zeolite T can coexist with other species, which develops either depending on time or temperature variations. Table 2 has a summary of the findings of their study.

Zeolite T can be developed at 100 °C only if it is conducted at 168 to 216 h, disappearing if the temperature increases to ≥ 120 °C [29]. Samples synthesized at 120 °C in a time of 144 h had no crystalline phase, while as the temperature increased to 150 °C, zeolite T peaks subdued and increased steadily as the temperature of 180 °C was applied. However, at 180 °C, crystallization of zeolite W was also observed (2T = 19.8 and 24.3) (Figure 10). At 130 °C, zeolite T with high purity was synthesized in 144 h, while restricting the formation of other species like zeolite W.

Table 3 shows the effect of different synthesis methods as well as the temperature and time towards the crystallization of zeolite T. Different methods were employed, namely (i) conventional refluxing heating (CR), (ii) conventional hydrothermal heating (CH), (iii) microwave refluxing heating (MR) and (iv) microwave hydrothermal heating (MH) [23]. As the synthesis temperature remains constant at 100 °C, the crystallization behavior varies in all synthesis techniques. The result shows that the lowest time of 48 h did not provide sufficient conditions to impute the formation and growth of zeolite T nuclei. However, as the zeolite T crystals form at 120 h, the crystallization remained, even though synthesis time was prolonged to 168 h.

Therefore, at a favorable temperature of 120–130 °C, increasing and prolonging the crystallization time does not significantly affect the crystallinity of zeolite T [23,29]. This suggests that 120 h is the minimum time needed to develop and grow zeolite T, via the hydrothermal method.

### 2.5. Effects of the Addition of SDA

Structure directing agent (SDA) is the substance implanted into the composition gel when synthesizing any desired zeolite. It acts as a template base and will be dismissed right after the crystallization phase. There are different SDA types: tetramethylammonium hydroxide (TMAOH), TMPBr_2_, tetraethylammonium hydroxide (TEAOH), to name a few. The presence of SDA in the synthesis system carries a significant role, as it determines product crystals’ impeccability. Moreover, SDA can accelerate the nucleation and crystallization phase. As an example, for TMAOH, the hydroxide anion within it will provide the necessary alkalinity to facilitate the formation of anionic silicates, thus expediting the rate of dissolution of silicate and aluminate [3,7,20,21,28,30]. Additionally, the authors in [7] reported that providing an abundant amount of SDA will create high supersaturation conditions and steric stabilization of proto-nuclei, which are the critical factors in forming zeolite nanoparticles without aggregation [30]. Determination of the exact amount of SDA needed in synthesizing zeolite, especially ERI-type zeolite, is crucial because erionite (ERI) intergrowth crystals will not form effectively if the system contains a large amount of cation [7,13,20,21].

In order to determine the effect of SDA upon crystallization behavior of T-type zeolite, a different amount of TMAOH was added into the mixture system of 1 SiO_2_: 0.025 Al_2_O_3_: 0.15 Na_2_O: 0.15 K_2_O: *x* TMAOH: 14 H_2_O, (*x =* 0, 0.01, 0.02, 0.04, 0.06, 0.1), where it was determined that under zero SDA, no crystalline phases are formed up to 192 h. Aided by XRD spectrum analysis, the addition of SDA at a temperature of 120 °C demanded 0.06 of TMAOH to synthesis zeolite T, with a particle size of 500–700 nm, where it showed the maximum intensity peak (2 theta = 7.7°) and gradually decreased as more TMAOH was added.

When a different ratio of TMAOH was added into the composition gel of 1 SiO_2_: 0.055 Al_2_O_3_: 0.23 Na_2_O: 0.008 K_2_O: *x* TMAOH: 14 H_2_O, where *x* = 0, 0.05, 0.10, 0.15 and 0.22, it resulted in different crystallization behavior [3,21,24]. Synthesized zeolite T, without a TMAOH template, took 192 h to develop 5 µm length in 100 °C, while at 0.05 TMAOH, pure zeolite T started to develop as early as 48 h. Up to 0.15, 1–2 µm, rod-like crystals of zeolite T were obtained as pictured in Figure 11. However, at 0.22 TMAOH, the rod-like shape crystals deformed into small irregular particles, which confirmed that zeolite T with nano-sized crystals could be prepared in 0.22 TMAOH. Therefore, the varied additions of TMAOH amount into the synthesis gel can prepare zeolite T aggregates of a size range from 2 µm to 200 nm. Though the addition of SDA is proven to enhance the crystallization, it is crucial that alkaline cation content in the system is maintained at a low level to prevent the aggregation of the negatively charged aluminosilicate subcolloidal particles [22,30].

The concept of the participation of SDA in zeolite T crystallization is due to the offretite/erionite-type zeolite’s favorable condition, which is typically developed in the presence of tetramethylammonium ion or NCOTs, nitrogen-containing organic templates, namely benzyltrimethylammonium (BTMA) cations, and 1,4-diazabicyclo-2,2,2-octane (DABCO) cations [7]. Even though the synthesis of offretite/erionite tends to develop in the presence of proper NCOT, NCOTs are costly and consume high temperatures in templates-removal, which generally affect the structure of synthesized zeolite crystals. Hence, NCOTs are avoided in the crystallization of zeolite crystals [7]. 

Therefore, for the past decade, TMAOH has been the only favored SDA parameter in developing offretite/erionite zeolites—zeolite T. TMAOH is actively involved in the templating of the crystallization of zeolite T, while potential alternatives to SDAs remain unfound.

## 3. Conclusions

In synthesizing the zeolite T, several aspects are taken into account to achieve the high purity of zeolite crystals. By using the conventional method of hydrothermal treatment, it is proven to produce zeolite T, with respect to SiO_2_/Al_2_O ratios, temperature, time, alkalinity, and pH, with or without SDA. However, if the synthesis is aided with assistance methods, such as microwave irradiation or sonication, the time taken to produce a similar quality of zeolite T is effectively less. Modification of zeolite T into a membrane has been possible via secondary growth and has improved the potential for further performances towards gas selectivity and permeability. Meanwhile, under certain conditions, zeolite T can develop and coexist with other zeolite phases (treated as impurities), as this phenomenon does affect the performance of zeolite T.

Finally, we hope that this review will contribute to the emergence of a new exploitation of zeolite T, both in synthesis and applications, which will provide extensive information about the interrelation on the evolution phase of zeolite T, L, and W, and more. 

## Figures and Tables

**Figure 1 materials-14-02890-f001:**
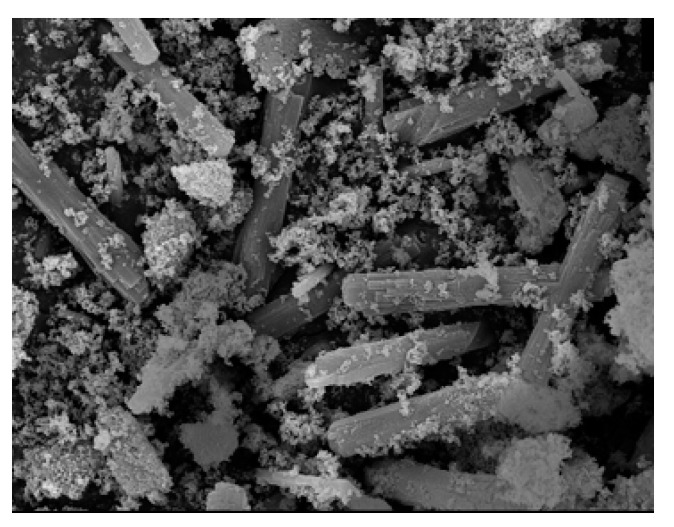
Rod-like shape of zeolite T synthesized by [3] (Reproduced with permission from Yin et al., Hydrothermal synthesis of hierarchical zeolite T aggregates using tetramethylammonium hydroxide as single template; published by Elsevier, 2015).

**Figure 2 materials-14-02890-f002:**
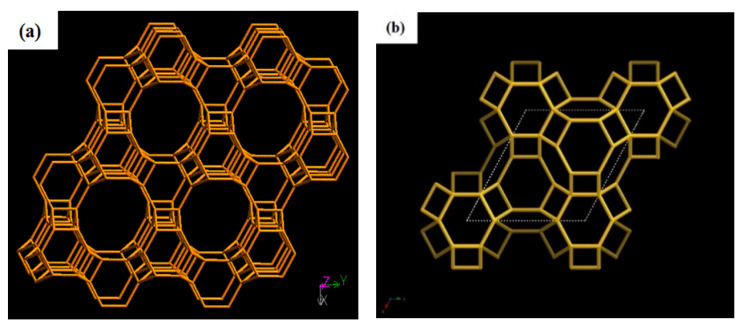
Framework of offretite (OFF) [001] (**a**) and erionite (ERI) [001] (**b**) (IZA).

**Figure 3 materials-14-02890-f003:**
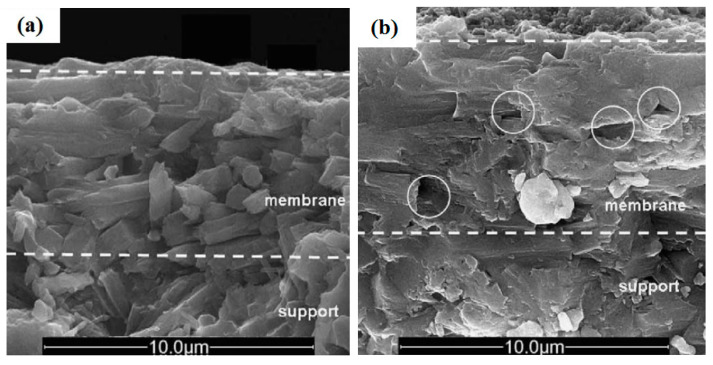
Cross-section of zeolite T membrane heated at 4.8 kW/m^3^ (**a**) and 6.0 kW/m^3^ (**b**); the circle indicates the area defects [11] (Reproduced with permission from Zhou et al., Microwave-assisted hydrothermal synthesis of a&b-oriented zeolite T membranes and their pervaporation properties; published by Elsevier, 2009).

**Figure 4 materials-14-02890-f004:**
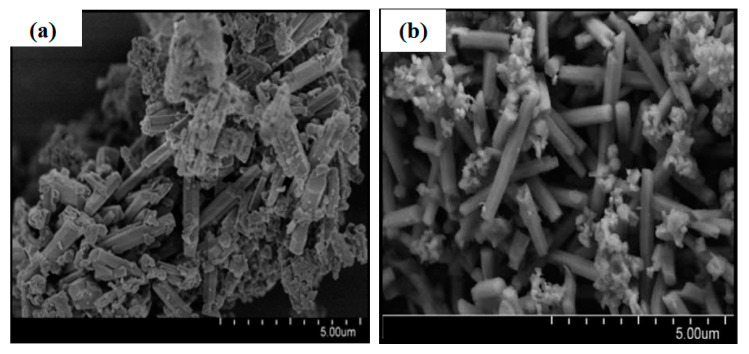
Morphology of zeolite T synthesized in 168 h without sonication (**a**), and zeolite T form in 24 h assisted with sonication pre-treatment (**b**) [12] (Reproduced with permission from Jusoh et al. Rapid-synthesis of Zeolite T via sonochemical-assisted hydrothermal growth method; published by Elsevier, 2017).

**Figure 5 materials-14-02890-f005:**
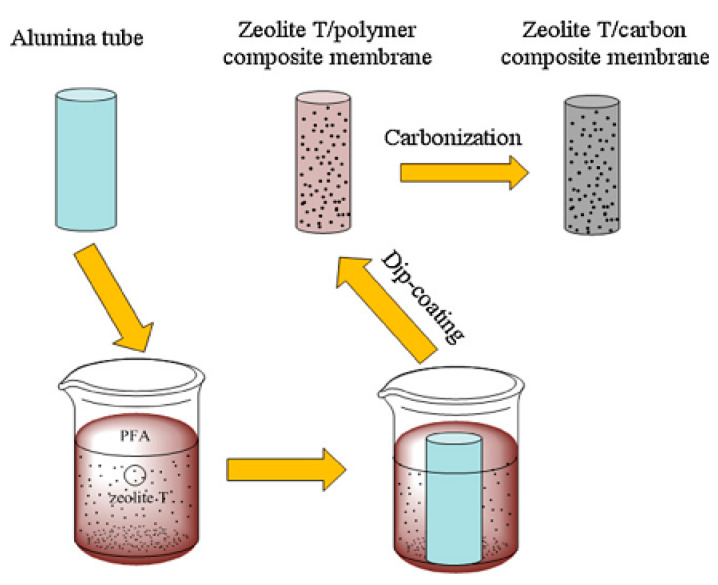
Schematic diagram of dip-coating method in preparing zeolite T/carbon composite [16] (Reproduced with permission from Yin et al., Thin zeolite T/carbon composite membranes supported on the porous alumina tubes for CO_2_ separation; published by Elsevier, 2013).

**Figure 6 materials-14-02890-f006:**
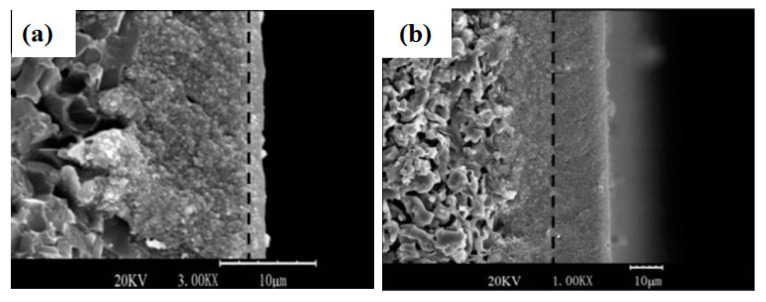
Morphology of zeolite T membrane undergone one coating cycle (**a**) and four coating cycle (**b**) [16] (Reproduced with permission from Yin et al., Thin zeolite T/carbon composite membranes supported on the porous alumina tubes for CO_2_ separation; published by Elsevier, 2013).

**Figure 7 materials-14-02890-f007:**
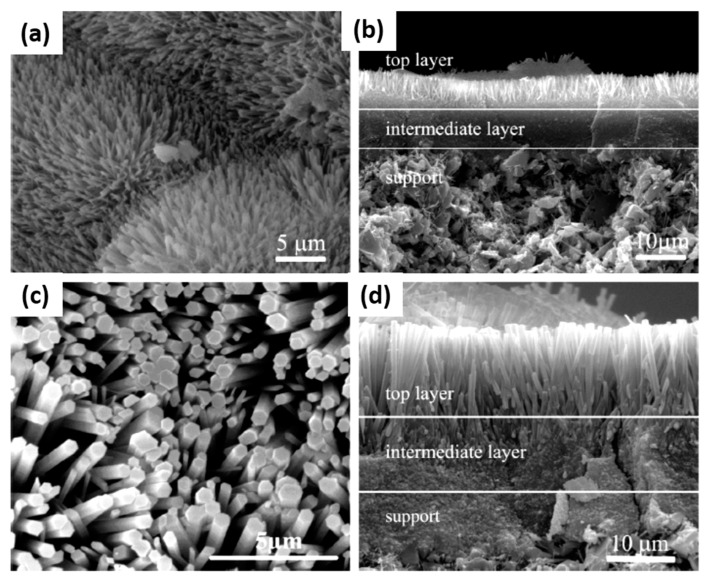
SEM for zeolite T membrane at 373 K (100 °C), 35 h for (**a**) and (**b**); 423 K (150 °C) for 20 h for (**c**) and (**d**) [5] (Reproduced with permission from Zhou et al., Synthesis of oriented zeolite T membranes from clear solutions and their pervaporation properties; published by Elsevier, 2013).

**Figure 8 materials-14-02890-f008:**
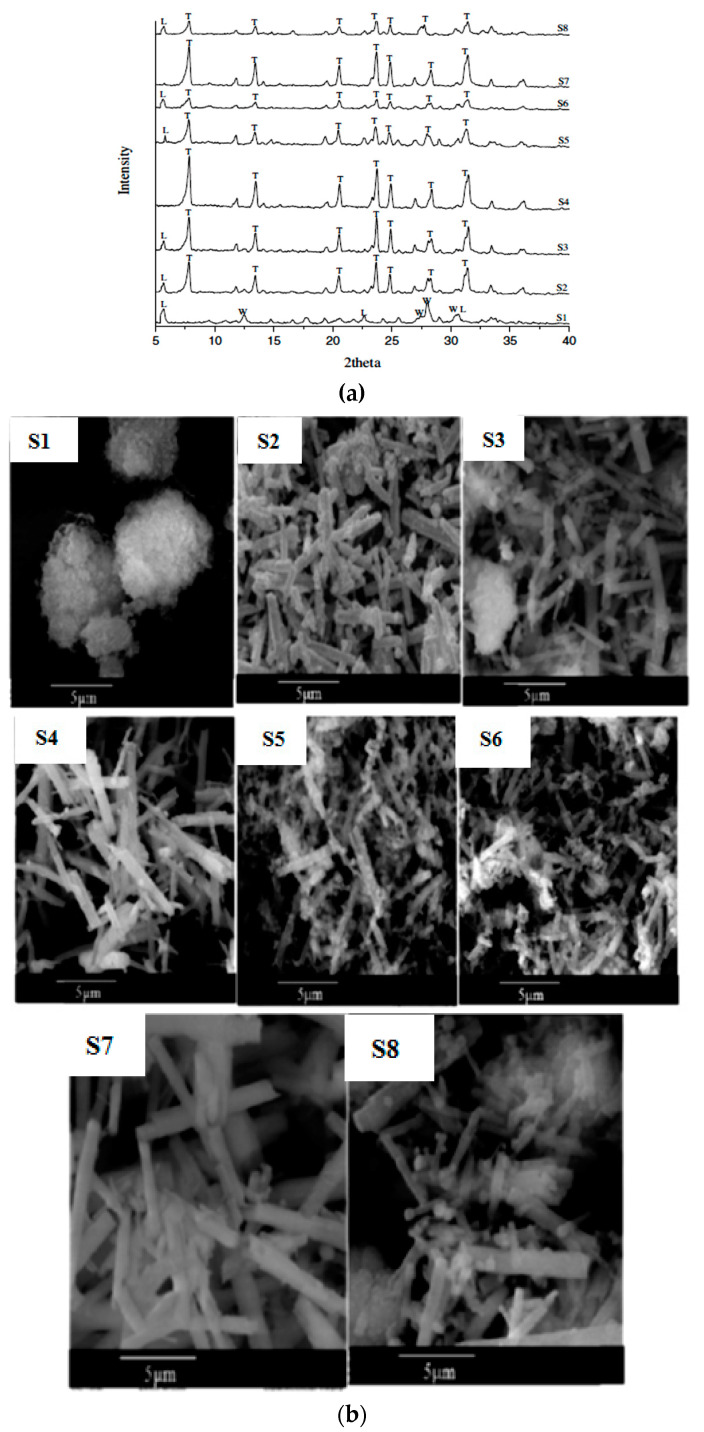
(**a**) XRD spectrum for sample S1-S8 [2] (**b**) SEM morphologies for sample S1–S8 [2] (Reproduced with permission from Rad et al., Development of T type zeolite for separation of CO_2_ from CH_4_ in adsorption processes; published by Elsevier, 2012).

**Figure 9 materials-14-02890-f009:**
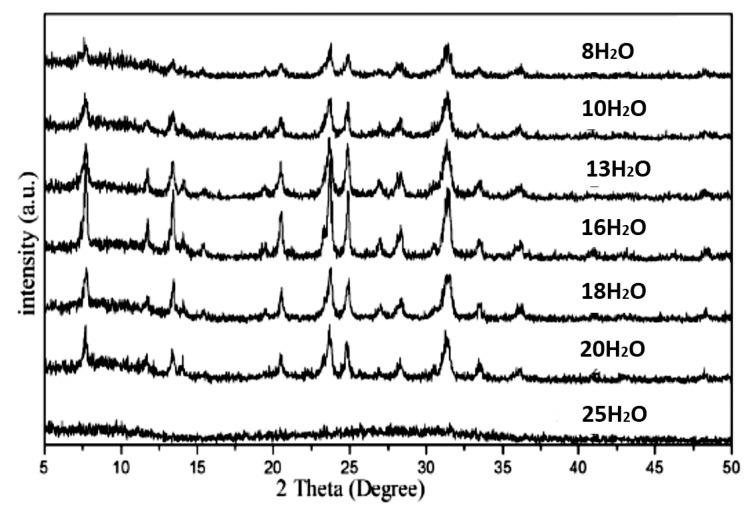
XRD spectrum or zeolite T’s samples synthesized at different moles of water [3] (Reproduced with permission from Yin et al., Hydrothermal synthesis of hierarchical zeolite T aggregates using tetramethylammonium hydroxide as single template; published by Elsevier, 2015).

**Figure 10 materials-14-02890-f010:**
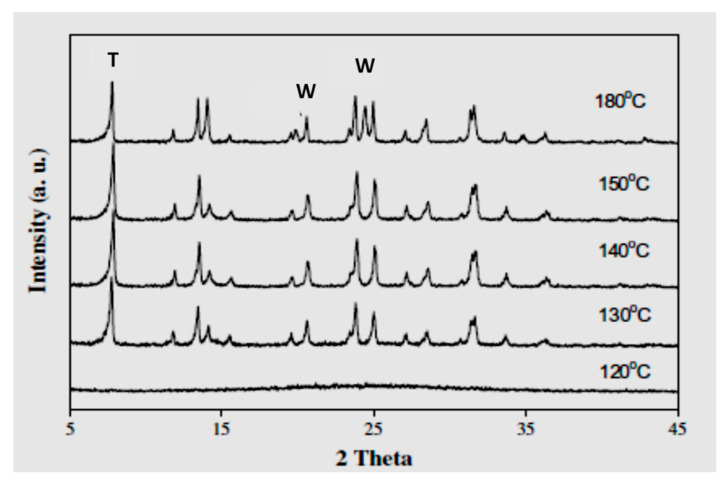
XRD analysis for zeolite T samples synthesized at different temperature for 144 h with formulation of 1 SiO_2_: 0.025 Al_2_O_3_: 0.15 Na_2_O: 0.15 K_2_O: 14 H_2_O: 0.06 TMAOH, where T and W indicate peaks of zeolite T and W, respectively [29] (Reproduced with permission from Jiang et al., Synthesis of T-type zeolite nanoparticles for the separation of CO_2_/N_2_ and CO_2_/CH_4_ by adsorption process; published by Elsevier, 2013).

**Figure 11 materials-14-02890-f011:**
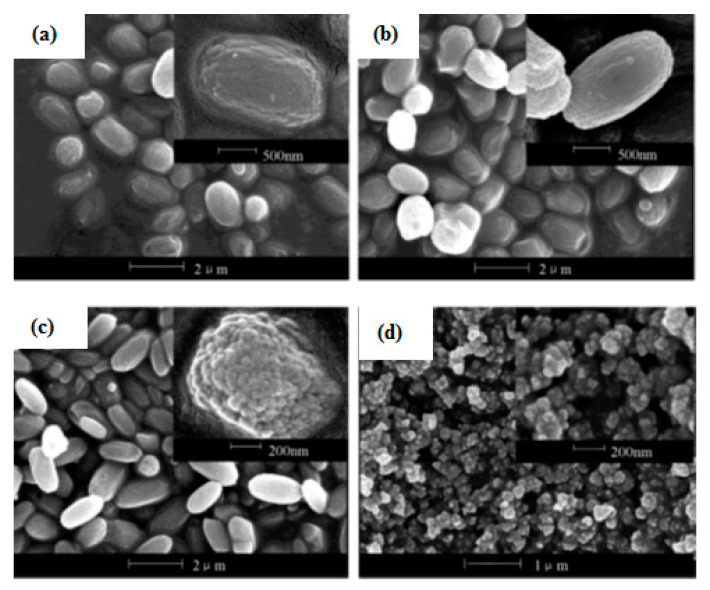
SEM images of zeolite T at 373 K (100 °C) at different amounts of TMAOH: (**a**) 0.05, (**b**) 0.10, (**c**) 0.15 and (**d**) 0.22 [3] (Reproduced with permission from Yin et al., Hydrothermal synthesis of hierarchical zeolite T aggregates using tetramethylammonium hydroxide as single template; published by Elsevier, 2015).

**Table 1 materials-14-02890-t001:** Effect of relative alkalinity toward product phases.

Sample	ηRM	α	Silica Sources	Relative Crystallinity	Product Phase
S1	20	0.82	Silica acid	0	L + W
S2	20	0.71	Silica acid	69	T + L
S3	25	0.71	Colloidal silica	80	T + L
S4	25	0.71	Colloidal silica	100	T
S5	25	0.82	Colloidal silica	54	T + L
S6	20	0.82	Colloidal silica	26	T + L
S7	20	0.71	Colloidal silica	82	T
S8	25	0.82	Silica acid	35	T + L

ηRM: SiO_2_/Al_2_O_3_ ratio, α: relative alkalinity. Source: [2] (Reproduced with permission from Rad et al., Development of T type zeolite for separation of CO_2_ from CH_4_ in adsorption processes; published by Elsevier, 2012).

**Table 2 materials-14-02890-t002:** Summary of evolution phases in synthesis zeolite T hydrothermally.

Sample	Crystallization Parameters		Obtained Product(s), Zeolite Type
	Temperature (°C)	Time (h)	
1	100	120	Majority: W
2	100	168	Majority: T
3	100	216	Majority: T, minor: W
4	120	120	Majority: T
5	120	168	Majority: T
6	120	216	Majority: W, minor: L
7	140	120	Majority: W, minor: L
8	140	168	Majority: L, minor: T, W
9	140	216	Majority: L

Source: [1] (Reproduced with permission from Mirfendereski et al., Investigation of hydrothermal synthesis parameters on characteristics of T type zeolite crystal structure; published by Elsevier, 2011).

**Table 3 materials-14-02890-t003:** Different synthesis methods and parameters affecting the crystallization behavior of zeolite T.

Experiment	Technique	Time (h)	Crystallization Behavior (XRD)
1	CR, T = 100 °C, P = 1 atm	48	Amorphous
2		72	Amorphous
3		96	T + amorphous
4		120	T
5		168	T
6	MR, T = 100 °C, P = 1 atm	5	Amorphous
7		10	Amorphous
8		20	Amorphous + T
9		30	T
10		48	T
11	CH, T = 100 °C, P = 1 atm	48	Amorphous
12		72	Amorphous + T
13		96	T
14		120	T
15		168	T
16	MH, T = 100 °C, P = 1 atm	5	Amorphous
17		10	Amorphous
18		15	Amorphous + PHI
19		20	PHI
20		30	PHI

Source: [23] (Reproduced with permission from Zhou et al., Synthesis of zeolite T by microwave and conventional heating; published by Elsevier, 2013).

## Data Availability

Not applicable.

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
