# Peer review of "Important Synthesis Parameters Affecting Crystallization of Zeolite T: A Review"

_materials, 2021, doi:10.3390/ma14112890_

Round 1
Reviewer 1 Report
T-zeolite represents intergrowth system of two pure zeolitic structures -
Erionite (ERI) an Offretite (OFF). Both consist from the same cancrinite units in chains, but in erionite each second cancrinite unit is turned by 60 °. This fact results in little different XRD spectra (Erionite has in addition 4 more reflexes in comparison with Offretite), and great difference in porous structure. Pore size of 8 member ring channels of 0.36 x 0.51 nm includes Erionite into group of narrow-pore zeolites, while pore size of 12 member ring channels of 0.67 x 0.68 ranks Offretite into wide-pore zeolites. It means that Erionit and Offretite significantly differ in sorption and diffusion properties and texture characteristics.
T-zeolite (in fact, mixture of Erionite and Offretite in certain ratio, which is not always the same) was in many cases the product of non-fully successful effort for synthesis of pure Erionite (synthesis of pure Offretite is more easily).
In fact, T-zeolite has no up today any important industrial catalytic application. The only potential use of Erionite in catalysis was possibility in selective cracking of n-alkanes in gasoline (Selectoforming), but Erionite was replaced by zeolite ZSM-5.
The most of references cited in the submitted paper deals with the preparation of T-zeolite as membrane for different gas-mixture separation. For this application, the most important properties are just texture, sorption and diffusion characteristics.
In review, some comparisons of XRD-spectra are shown, and many SEM – pictures (often poor quality) of synthesized materials. Authors in chapter 2 describe factors affecting crystallization of zeolite T, and the quality of products compare on the basis of XRD spectra and SEM images. For use of T-zeolite in mixture of gas separation (e.g. CO2/CH4) there must be taken into account texture, sorption and diffusion properties. Because T-zeolite is not pure zeolite, but always some intergrowth mixture of two pure zeolite structures ERI/OFF, the quantity and the distribution of ERI and OFF units, that influence the sorption and diffusion properties, depends on the methods of synthesis. But, in the review, anything about these most important properties in connection with the synthesis conditions is not mentioned.
The conclusion is very general.
Author Response
The SEM image has been improved and updated to recent study, and additional sub-chapter added in chapter 1.
As for the influence factors toward the application performance of zeolite T are generally included because this review was intend to discuss about important factors that need to be consider if any researcher interest to synthesis zeolite T.
The conclusion part has been rephrase.
Reviewer 2 Report
The present review paper is intended to describe the effect of various synthesis parameters on the formation of zeolite T. However, the article has been written in such a way that makes it very difficult to understand the reported information.
1) First, there are numerous vague or unclear formulations/sentences.
2) Second, the manuscript must be checked for spelling and grammar mistakes. The text must be rephrased correctly to allow the reader to catch the significance of its content. Generally speaking, the English used is incorrect and needs extensive editing. It is hard to understand the meaning of most of the phrases.
3) Third, several cited references are not listed in the section "References", such as "Flanigen et al, 1986", "Park et al", "Carmona et al", "Veerapur et al, 2008", "Cejka et al, 2007". Different styles of presentation of the references have also been used, so that the text does not read very smoothly.
4) The three main synthetic routes of zeolite T have been reviewed in the first part of the manuscript. This section needs to be more extensively discussed since many explanations are relatively superficial. No information is available on the gel composition, nature of the silica source, etc. The influence of the microwave irradiation time and power on the zeolite characteristics during the MW-assisted synthesis method have not been discussed too.
Regarding the second part of the manuscript, here is a non-exhaustive list of unclear points.
5) Page 8, lines 183-186: This sentence is too general. Please describe the corresponding data more in depth.
6) Page 8, lines 187-188: The meaning of this sentence is unclear. Please clarify what you mean by "determined on the crystallization behavior that has been applied".
7) Page 11, line 236: It is stated that a high alkalinity (of 0.71) is required for the synthesis of zeolite T, whereas on page 9 (line 217), the authors have specified "low optimum alkalinity of 0.71". Please clarify and/or correct the sentence.
8) Page 11, lines 241-242: What do the authors mean by "pH values range (0.71-0.82) which correspond to the values of water precursors of the solutions"?
9) Page 11, lines 243-244: What is the meaning of "mole of 8-16 fused in"? It is completely incomprehensible.
10) Page 13: The authors are referring to ambient temperature of 100°C and 120°C, which is a non-sense. Please make the required correction.
11) Page 13, line 283: "unexceptionally crystallization temperature and time". What does it mean??
12) Page 13, last paragraph: Please explain more in depth the influence of the synthesis method (CR, CH, MR, MH) on the zeolite phase obtained.
13) Page 16, conclusion, lines 359-360: It is unclear what is meant by "alteration of silica sources” and by "time factor that can be altered by providing more source of nucleation (silica sources)". Please add more detailed information.
I strongly recommend the authors to improve the scientific quality of their manuscript and to reformulate the text with a view to providing a more readable and unambiguous version.
Author Response
1) All unclear formulations has been standardized.
2) Grammar and spelling checked.
3) Citation and references format has been standardized. .
4) Additional sub-chapter added in chapter 1. Additional explanation on how microwave or sonication helps to impute the crystallization of zeolite T also added.
Regarding the second part of the manuscript, non-exhaustive on unclear points has been elucidated. Some aspect remain unchanged because the depth of this review is to list all the important factors that affecting the synthesis of zeolite T and quality of the product (with or without competitive species) only.
Reviewer 3 Report
The present review aims at summarizing the information on the synthesis parameters influence the crystallinity of the T zeolite. The T zeolite constitutes a disordered intergrowth of Eri and OFF structures (where the latter prevails). The T zeolite has outstanding properties that can be used in catalysis and separation science, etc. There is an interest in the synthesis of the material and thus the review as the present one should be welcome by the scientific community. One may expect the article will be highly cited as well. However, I cannot recommend publishing the present form of the article. There are two reasons for this. First of all, the article contains a large number of language errors which affect the understanding of the presented material. The text should undergo a profound English editing before being revised for the second time. The second major issue is the fact it lacks some recent advancements in the T zeolite synthesis. In order for the article to be more credible Authors should once more check the literature data bases for the articles dealing with the T zeolite synthesis. In the following I include some of missing references:
Yin, X.; Chu, N.; Lu, X.; Li, Z.; Guo, H. Cost-Effective Two-Stage Varying-Temperature Rapid Crystallization of Zeolite T and SAPO-34. J. Cryst. Growth 2016, 441, 1– 11, DOI: 10.1016/j.jcrysgro.2016.02.003
Jusoh, N.; Yeong, Y. F.; Mohamad, M.; Lau, K. K.; Shariff, A. M. Rapid-Synthesis of Zeolite T via Sonochemical-Assisted Hydrothermal Growth Method. Ultrason. Sonochem. 2017, 34, 273– 280, DOI: 10.1016/j.ultsonch.2016.05.033
Jing Liu, Jiayou Zhang, Huizhi Zhang et al “ Synthesis of hierarchical zeolite T nanocrystals with the assistance of zeolite seed solution” J Solif State Chem 285, 2020, Article Number: 121228
Finally, some recent reviews on zeolites synthesis should be cited as well.
Author Response
Improving the scientific aspect of this manuscript has been taken into action.
Missing references and recent study has been added.
Round 2
Reviewer 2 Report
The revised version of the manuscript has been greatly improved. Most of my concerns have been addressed. However, there are still some minor points that need to be checked before the article being accepted for publication.
1) References have not been listed throughout the text in an increasing order of citation. For example:
- Page 1: reference [5] is cited in line 35, then followed by reference [14] in Figure 1 caption.
- Page 3: reference [9] is missing. [8] is cited in line 61, then [10-12] in line 66.
- Page 4, line 97: reference [17] in §1.3 is not referring to the sonochemical-assisted method, but to the secondary growth method (§1.4).
- Page 4: Images presented in Figure 4 are not issued from reference [18].
- Page 5: reference [20] is missing. Reference [19] is cited in line 110, then [21] in line 120.
- Pages 6-7: reference [22] is cited in line 143, then reference [32] in line 159. Then [25] in line 160, followed by [31] in line 164.
Please check carefully all reference numbers throughout the text of the manuscript and list them in increasing order of citation.
2) Page 5:
Line 109: (> 4727 C) to be replaced by (> 5000 K).
Line 110: (> -166 C) to be replaced by (> 107 K s-1).
Line 119: -90-100°C to be replaced by 90-100°C (the first "-" sign to be removed)
3) Page 8:
Line 194: “alkalinityies” to be replaced by “alkalinities”.
4) Page 10:
Figure 8 caption: S1-S2 to be replaced by S1-S8.
Line 217 “the range of pH values (0.71-0.82)”: these are not values of pH.
So please replace (0.71-0.82) by the corresponding pH values or replace “range of pH values” by “range of alkalinity values”.
5) Page 16, line 345: “with elements such as …” to be replaced by “with assistance methods such as …”.
Author Response
All corrections have been carried out especially on the sequence of reference.
Reviewer 3 Report
The corrections made by Authors substantially improved the article. There are still, however, minor issues which should be clarified.
On page 8, lines 183-184 one reads: "Generally, when the medium is at a high 183 alkalinity level of about 0.71 [2], it will accelerate the nucleation" - please define "alkalinity, as this quantity is not commonly used in ge neural chemistry. A similar term, relative alkalinity, is used also in Table 1. Please specify. This is misleading, because in lines alkalinity is probably mixed up with pH ("pH values (0.71- 0.82)").
Statements regarding the temperature of synthesis are contradictory. According to the information in line 267, the T phase disappears above 120 °C, according to the information in line 284-285 the optimum temperature is in between 120 and 130 °C. If literature data are partially contradictive, please state this clearly.
Author Response
Comment 1
Line 183-184
Correction: relative alkalinity with respect to water
Comment 2
Line 284-285
Zeolite T can be developed at 100°C if only its conducted at 168 to 216 hours, disappearing if the temperature increases to ≥ 120°C [9]. Sample synthesized at 120°C in time of 144 h has no crystalline phase while as the temperature increased up to 150°C, zeolite T peaks are subdued and increased steadily as the temperature applied at 180°C. However, at 180°C, crystallization of zeolite W is also observed (2T = 19.8 and 24.3) (Figure 10). At 130°C, zeolite T with high purity was synthesized in 144 h, while restricting the formation of other species like zeolite W.
Explanation:
At 120 °C (144 h), there is no crystallization of zeolite T observed but it did not indicate the crystals subdued because at this temperature the nucleation is happening but not sufficient enough to allow the crystals of zeolite T to grow.
At 130 °C (144 h), shows the positive growth of zeolite T crystals, however as the temperature increased up to 150 °C, the zeolite T crystals subdued because of the co-exists properties of this species. As the Zeolite W crystals grow, the percentage of zeolite T subdued also increases.
Why did we choose to state 120°C – 130 °C is the optimum range for synthesis temperature, it is because, in this journal, there were several range of synthesis time used to synthesis zeolite T at different temperatures. At 144 h, no crystallinity at 120 °C, but as the time increased to 120 °C, the nucleation phase turns into crystallization.
It is important to consider the strong connection between time and temperature during the synthesis of this zeolite. If we want to synthesis zeolite T at 100 °C, we can – but we need to increase the synthesis time up to 168 h – 216 h. however, if we increase the temperature while maintaining the synthesis hour at 216 h, zeolite T will disappear due to the substitution of other species (Zeolite W or Zeolite L).